# Infrared Small Target Detection Algorithm Based on Improved Dense Nested U-Net Network

**DOI:** 10.3390/s25030814

**Published:** 2025-01-29

**Authors:** Xinyue Du, Ke Cheng, Jin Zhang, Yuanyu Wang, Fan Yang, Wei Zhou, Yu Lin

**Affiliations:** 1Kunming Institute of Physics, Kunming 650223, China; xinyuedu_211@163.com (X.D.); zhangjin_211@163.com (J.Z.); wxyjin232425@163.com (Y.W.); yf7334@163.com (F.Y.); 2National Pilot School of Software, Engineering Research Center of Cyberspace, Yunnan University, Kunming 650091, China; chengke@stu.ynu.edu.cn (K.C.); zwei@ynu.edu.cn (W.Z.)

**Keywords:** infrared small target detection, attention mechanism, bottom-up feature fusion

## Abstract

Infrared weak and small target detection technology has attracted much attention in recent years and is crucial in the application fields of early warning, monitoring, medical diagnostics, and anti-UAV detection.With the advancement of deep learning, CNN-based methods have achieved promising results in general-purpose target detection due to their powerful modeling capabilities; however, CNN-based methods cannot be directly applied to infrared small targets due to the disappearance of deep targets caused by multiple downsampling operations. Aiming at these problems, we proposed an improved dense nesting and attention infrared small target detection method based on U-Net called IDNA-UNet. A dense nested interaction module (DNIM) is designed as a feature extraction module to achieve level-by-level feature fusion and retain small targets’ features and detailed positioning information. To integrate low-level features into deeper high-level features, we designed a bottom-up feature pyramid fusion module, which can further retain high-level semantic information and target detail information. In addition, a more suitable scale and position sensitive (SLS) loss is applied to each prediction scale to help the detector locate the target more accurately and distinguish different scales of the target. With our IDNA-UNet, the contextual information of small targets can be well incorporated and fully exploited by repetitive fusion and enhancement. Compared with existing methods, IDNA-UNet has achieved significant advantages in the intersection over union (IoU), detection probability ($P_{d}$), and false alarm rate ($F_{a}$) of infrared small target detection.

## 1. Introduction

Accurate target identification is a crucial technological foundation for effective Infrared Search and Track (IRST) operations. It enables the early detection of targets and provides precise warnings, which are critical requirements for air defense systems. However, the targets in infrared images are often characterized by their small size or significant distance from the detector, posing challenges to reliable detection. In images captured by electro-optical (EO) systems, such targets typically occupy only a few or even a single effective pixel, or in some cases, they may appear as sub-pixel features. According to the SPIE international standards, targets that occupy less than 0.12% of the image size are classified as small targets. These targets usually appear as dots or spots with minimal texture or discernible shape features [1]. Moreover, due to atmospheric scattering and absorption, infrared radiation further weakens with distance, making the target’s brightness exceedingly faint. Recent academic studies have shown an increasing focus on infrared faint and sub-pixel target discrimination, highlighting its growing importance in advancing detection methodologies.

In target detection, researchers have proposed several traditional methods in early studies in the field, including filter-based algorithms [2,3], local contrast-based methods [4], and image data structure-based methods [5,6]. However, these conventional approaches predominantly rely on target intensity profiles and local neighborhood characteristics for detection, with their performance constrained by the inherent limitations of manual feature engineering.

The current detection landscape has transitioned from manual feature design to convolutional neural networks (CNNs) as the principal deep learning framework [6,7,8,9]. As the predominant deep learning framework, CNNs have demonstrated remarkable capabilities in extracting robust feature representations, particularly in challenging weak target detection scenarios. Leveraging CNN’s ability to process large-scale data and its powerful model-fitting capabilities, these methods can effectively capture discriminative and salient features of small targets. Typically, these methods focus on the target itself, utilizing multi-layer networks to extract feature information of the target for detection. Enhancing detection accuracy is often achieved by increasing the depth of the neural network. However, it is worth noting that since small targets in images occupy only a few effective pixels, and given the significant differences in shape and texture between visible and infrared targets, infrared small targets lack texture and structural information compared to visible light. Therefore, simply deepening the network does not necessarily improve the detection performance for small targets [10]. On the contrary, due to the scarcity of features in infrared small targets and the limited scale of available datasets, the features of small targets face the risk of being overwhelmed by background features during the detection process after multiple down-sampling operations.

To solve these problems, we proposed an improved dense nesting and attention infrared small target detection algorithm based on U-Net (IDNA-UNet). Our framework utilizes a densely connected interaction network as the core computational backbone to facilitate multi-level feature abstraction from infrared image inputs. In order to avoid using too many pooling operations, we only use a five-layer network structure to expand the receptive field while extracting features. Through progressive feature fusion and interaction, the intermediate nodes can receive the features of themselves and the adjacent two layers of nodes, thereby fully utilizing and integrating the contextual information with small target information. The extracted features include deep features with high-level semantic information and shallow features with rich image contour and grayscale information. The weights of feature channels in traditional convolutional neural networks are the same. However, during the convolution transformation of feature information, the feature areas that each channel should focus on in the infrared image are different, which requires focusing on channels with small target information. Two architectural innovations are implemented: a channel attention module to enhance target-relevant feature channels, and a multi-scale fusion mechanism employing a bottom-up pyramid architecture for optimized feature integration across different levels. The advantage of this is that it can protect the representation of deep, small targets while extracting high-level features. Since the widely used intersection-over-union (IoU) loss lacks sensitivity to target size and position, it is difficult to distinguish targets of different scales and positions for target detection tasks, which greatly limits the performance of the detector. A novel and effective scale- and position-sensitive loss function is used to improve the existing problems. In summary, the contributions of this paper are summarized as follows:Based on the encoder and decoder architecture of U-Net, we proposed an improved densely nested and attention infrared dim small target detection network architecture. By introducing a densely nested interactive network as the backbone network, we integrated a channel attention mechanism to enhance feature representation by adaptively weighting different channels during infrared dim small target feature extraction.Through the architectural design of multi-scale heads and bottom-up feature pyramid feature fusion, we can deepen the focus on target features layer by layer. This structural innovation substantially augments the network’s capacity for detecting subtle features and effectively distinguishing infrared targets against cluttered backgrounds.A novel and effective scale-sensitive and position-sensitive loss function is adopted, which helps the detector to distinguish objects of different scales and positions, thereby further improving the detection performance.

## 2. Related Work

Top-hat [11] is a classic morphological filtering method that uses various filters to estimate the background of a scene, allowing the target to be distinguished from complex backgrounds. These algorithms exhibit significant dependence on target–background size differentials, consequently constraining their practical application scope. Gao C et al. [5] innovatively expanded the conventional infrared image framework through the development of an Infrared Patch-Image (IPI) model, achieved via localized block construction methodology. By vectorizing and concatenating overlapping image patches sampled using a sliding window, they created a matrix that satisfies the assumptions of low-rank background and sparse targets. PSTNN [12] employs a non-convex method based on the tensor nuclear norm, while RIPT [6] focuses on reweighted infrared patch tensors. Subsequently, other scholars proposed methods based on visual mechanisms. For instance, the research team led by Chen et al. [4] proposed the Local Contrast Measurement (LCM) framework, employing localized contrast analysis to differentiate targets from background elements. The limitations of these algorithms lie in their sensitivity to scene changes, as they struggle with parameter and algorithm adjustments, making them unsuitable for multi-scene applications. Moreover, traditional algorithms face an inherent trade-off between improving sensitivity and reducing false alarm rates, which is difficult to reconcile.

With the rise of deep learning, methods leveraging the massive data and powerful fitting capabilities of CNNs have demonstrated better performance compared to traditional methods. However, general object detection algorithms perform poorly in detecting infrared small and weak targets. The challenges lie in the conflict between the target’s size and the deep feature extraction network, as well as the poor generalization ability of the algorithms. To address this problem, researchers have proposed some improvement and optimization strategies based on the existing target detection methods.

To enable the network to learn infrared small target features across different scales as effectively as possible, Dai Y [13] and others proposed an Asymmetric Context Modulation (ACM) method to enhance features. Zhang T [14] further extracted contextual information of the target in deeper layers of ACM. This architectural module is specifically designed to generate multi-scale feature representations through the implementation of dilated convolutions and adaptive global average pooling operations on high-level feature maps. Generative adversarial networks (GANs) achieve a Nash equilibrium through adversarial games between the generator and the discriminator. In the task of infrared small target detection, Wang H et al. [8] pioneered a novel approach for small target detection through the development of an enhanced generative adversarial network architecture. The researchers conceptualized infrared small target segmentation as an optimization challenge, focusing on achieving equilibrium between missed detection (Md) and false alarm (Fa) rates through balanced parameter optimization. Zhao et al. [15] trained the generator using simulated targets and then utilized the trained generator for target detection. Zhou et al. [16] established the PixelGame framework, reformulating multi-scale infrared target detection as a competitive optimization problem with adversarial balancing of detection errors. Concurrently, Li et al. [9] and collaborators introduced DNANet, implementing dense nested attention mechanisms for improved detection performance. The densely nested interactive architecture of the backbone network facilitates comprehensive contextual information integration through iterative fusion and enhancement processes, optimizing small target feature utilization. He et al. [17] designed the Subpixel Sampling Cuneate Network (SPSCNet) to achieve iterative fusion between high-level and low-level features. UNet-based methods have demonstrated remarkable progress across various segmentation tasks. Wu et al. [10] proposed UIU-Net, which integrates a compact U-Net [18] within a larger U-Net backbone to simultaneously capture global and local contrast information. Similarly, Zhang et al. [19] introduced Sharp U-Net in 2022, leveraging depthwise separable convolutions to optimize lightweight network design, achieving substantial reductions in both parameter count and computational complexity while maintaining competitive segmentation accuracy. While these architectures have shown exceptional performance in biomedical image segmentation, particularly in terms of inference speed and resource efficiency, their applicability to infrared small target detection remains limited due to unique challenges such as target sparsity and complex feature representation.

With the active exploration of various researchers, deep network frameworks specifically designed for infrared small target detection have been continuously proposed, leading to progressively improved detection performance. However, existing methods still face several challenges, such as limited feature representation capabilities, insufficient robustness against complex backgrounds, and constraints imposed by limited network depth that hinder further performance enhancements. These issues restrict the effectiveness and applicability of infrared small target detection in real-world scenarios. To address these limitations, this paper aims to enhance the feature representation capability of each layer within a limited network depth to further improve detection robustness, which is crucial for the performance of infrared small target detection.

Infrared images have a more extreme contradiction between resolution and semantics. On the one hand, due to the long imaging distance, the proportion of small infrared targets in the image usually only occupies a few pixels, and its features are more likely to be covered by the surrounding background features as the number of network layers increases. On the other hand, in many real scenes, small infrared targets are often buried in complex backgrounds with relatively low signal clutter, lacking corresponding texture and shape features. However, deep convolutional neural networks usually use the method of gradually reducing the size of the feature maps to increase the receptive field so as to learn more semantic information. The resolution and semantic level of their feature maps are often a contradiction. Therefore, within constrained network depth configurations, maximizing the representational capacity of features at each network layer becomes critical for enhancing infrared weak and small target detection performance. The above methods still have some shortcomings. Directly using these methods may cause the loss of deep, small targets. Consequently, considerable scope for algorithmic refinement persists in deep learning-based solutions for infrared weak target detection applications.

## 3. Methodology

This section introduces IDNA-UNet, including its overall structure, core module structure, and the design of the SLS loss function.

### 3.1. The Overall Structure

IDNA-UNet consists of two parts: feature extraction and feature fusion. The feature extraction module is inspired by the nested structure in the field of image segmentation and the attention mechanism in the field of object detection [9]. The feature fusion module designs a bottom-up feature pyramid fusion module (BFPFM) to integrate low-level features into deeper high-level features.

In the multiple down-sampling stages of infrared small target detection, high-dimensional feature representations, despite containing target semantic data, frequently fail to effectively maintain and accentuate the critical attributes of low-visibility small targets. On the other hand, low-dimensional features can effectively retain target characteristics but lack high-level semantics. This represents a significant bottleneck in infrared small target detection. Therefore, the existence of this inherent contradiction should be carefully considered at the outset of network design, and targeted improvements to the network architecture should be made. As shown in Figure 1, IDNA-UNet is based on the encoder and decoder architecture of U-Net. Our backbone network structure retains the dense nesting structure (DNIM) of DNANet [9], which adds multiple skip connection paths based on U-Net to achieve hierarchical feature fusion, which effectively utilizes the global contextual information of small targets. Compared to DNANet, we introduce the SENet module at the end of each convolutional neural network. In traditional CNNs, pooling layers tend to weaken the target features within each sample, while SENet reallocates the weights based on the importance of each channel of features, effectively enhancing the weights of the channel information. The feature maps $x_{i}$ of different scales are output from the backbone network and $x_{i}$ is input into a feature fusion module with multi-scale heads. Multi-scale feature representations are normalized to uniform dimensionality through systematic upsampling operations. A scale and position sensitive (SLS) loss is applied to ensure that each scale head can generate predictions of the corresponding scale. The predictions of different scales are connected for the final prediction.

### 3.2. Feature Extraction Network

In order to ensure high-resolution features and fine positioning information, the feature extraction module designs a dense nested interactive module (DNIM) based on the U-Net framework. On the basis of the U-Net architecture, multiple jump connection paths are added between the encoder and the decoder. The sub-encoder receives the detailed positioning information of the encoder through the jump connection path and directly applies the shallow position information to the deep feature map, thereby maintaining the detailed information of the target. In addition, the output of the shallower sub-encoder node will be connected to the deeper sub-encoder node of the same size to utilize the detailed features further.

$X^{i,j}$ represents the current node output, where *i* represents the index of the downsampling layer and *j* is the index of the dense block convolution layer along the normal skip connection path. When j=0, each node only receives features from dense normal skip connections. The feature map represented by $X^{i,j}$ is calculated as(1)$$\begin{array}{c} X^{i,j}=P_{\max}\left(F\left(X^{i-1,j}\right)\right) \end{array}$$
where F(·) represents multiple cascaded convolutional layers of the same convolutional block and $P_{max}(\cdot )$ represents maximum pooling with a stride of 2. When j>0, each node receives output from three directions, including dense ordinary jump connections and nested interactive jump connections, and the feature map stack represented by $X^{i,j}$ is calculated as(2)$$\begin{array}{c} X^{i,j}=\left[F{\left[X^{i,k}\right]}_{k=0}^{j-1},P_{\max}\left(F\left(X^{i+1,j-1}\right)\right),u\left(F\left(X^{i-1,j}\right)\right)\right] \end{array}$$
where u(·) represents the upsampling layer, [·,·] represents a cascade layer, and k=0 indicates the $X^{i-1,0}$ convolutional block.

In the dense nested feature extraction module network, each $X^{i,j}$ represents a convolutional neural network. The feature extraction module (FEM) architecture implements a novel Residual SENet Block (RSB) node design, supplanting conventional ResBlocks. Each RSB node integrates ResNet18’s residual unit architecture with SENet’s channel attention mechanism, forming a hybrid structure that combines residual learning with adaptive channel-wise feature recalibration. Our design philosophy emphasizes two core objectives: (1) maintaining model lightweightness through parameter-efficient architecture design, and (2) achieving substantial global feature enhancement with minimal computational overhead. To preserve architectural consistency, each RSB node incorporates skip connections immediately following the initial convolutional layer, as illustrated in Figure 2.

Small targets can be easily submerged in the high-level features of the network, but using only low-level features, the semantic information cannot be grasped well, which can easily lead to missed detection and false alarms. The SENet framework employs residual-based learning to establish feature channel importance metrics, dynamically allocating weight values to enhance network focus on discriminative feature maps while suppressing less informative ones. Invalid or ineffective feature maps have small weights. SENet mainly includes two operations: squeeze and excitation.

In Figure 3, squeeze compresses the W×H×C feature map containing global information into a 1×1×C feature vector through global average pooling. This feature vector has a global receptive field and can be expressed as described below.

The first step of SENet is to extract the feature map of the image using the convolutional operation which can be represented as(3)$$\begin{array}{c} F_{tr}:X\to U,X\in {\,}^{H^{\prime }\times W^{\prime }\times C^{\prime }},U\in {\,}^{H\times W\times C} \end{array}$$(4)$$\begin{array}{c} F_{tr}:u_{c}=v_{c}*x=\sum_{s=1}^{c^{\prime }}v_{c}^{s}*x^{s} \end{array}$$

$F_{tr}$ represents a transformation operation that maps the input raw image *X* to the feature map *U*, where *H* and *W* denote the height and width of the feature map, and *C* represents the number of channels. $v_{c}^{s}$ represents the parameters of the *c* convolutional kernel, * denotes convolution, $x^{s}$ represents the *s* input, $V=[v_{1},v_{2},\ldots ,v_{c}]$ is the learned combination of convolutional kernels, and $U=[u_{1},u_{2},\ldots ,u_{c}]$ is the output.

In the squeeze stage, the size of the input feature *U* is H×W×C. Through a global average pooling operation, the features at each spatial location are aggregated into a single value, compressing each H×W region into a single value, resulting in a 1×1×C feature vector. It can be expressed as(5)$$\begin{array}{c} z=F_{sq}\left(u_{c}\right)=\frac{1}{HW}\sum_{i=1}^{H}\sum_{j=1}^{W}u_{c}(i,j) \end{array}$$

*z* represents the weights generated by the squeeze operation, and $F_{sq}(\cdot )$ represents the function of the squeeze operation.

The squeeze operation captures the global features of the feature channels, while the excitation operation captures the correlations between the feature channels, and is implemented using two fully connected layers. The initial fully connected layer implements dimensionality reduction, optimizing parameter efficiency while improving model generalization. Subsequently, the secondary fully connected layer reconstructs the original feature dimensions. The calculation process can be expressed as(6)$$\begin{array}{c} s_{c}=F_{ex}(z,W)=\sigma \left[W_{2}\delta \left(W_{1}z\right)\right] \end{array}$$
where $F_{ex}(\cdot )$ represents the function of the squeeze operation, δ denotes the ReLU function, and σ represents the sigmoid function. $W_{1}$ and $W_{2}$ represent the fully connected (FC) operations, and $S_{c}\in R^{1^{\times }1^{\times }C}$ denotes the generated attention weights.(7)$$\begin{array}{c} {\bar{X}}_{c}=F_{\mathrm{scale}}\left(u_{c},s_{c}\right)=s_{c}⊗u_{c} \end{array}$$
where ⊗ represents element multiplication, and ${\bar{X}}_{c}$ is the output feature map. $F_{scale}(u_{c},s_{c})$ refers to the per-channel multiplication between the weight vector $s_{c}$ and the original feature map $u\in R^{H^{\times }W}$.

### 3.3. The Feature Fusion Module of Multi-Scale Heads

After the feature extraction module, we designed a feature fusion module to aggregate multi-scale features. Let $x_{i}\in R^{H_{i}\times W_{i}\times C_{i}}$ be the feature map at the i-th scale of the DNIM decoder, where $H_{i}\times W_{i}$ is the spatial size, $C_{i}$ is the number of channels, and $i\in \left\{0,1,2,3,4\right\}$.

The i-th prediction $p_{i}\in R^{H_{i}\times W_{i}\times 1}$ is obtained through the corresponding prediction head, which is implemented by a convolutional layer followed by a sigmoid activation function(8)$$\begin{array}{c} p_{i}=Sigmoid(Cov(x_{i})) \end{array}$$

The prediction results $p_{i}$ from different scale heads are upsampled and input into the BFPFM module for further fusion.

### 3.4. Bottom-Up Feature Pyramid Fusion Module (BFPFM)

In a common convolutional neural network, as the number of network layers increases, the network’s ability to extract semantic features of the input becomes stronger. However, as the number of downsampling layers increases, the network is prone to lose spatial information on small-scale targets. The feature fusion architecture employs a bottom-up pyramid framework (BFPFM) to effectively encode and preserve small target characteristics by progressively integrating low-level spatial details into high-level semantic features.

BFPFM is shown in Figure 4. The local channel attention mechanism *L* locally aggregates the channel feature context of each spatial position, and the formula is(9)$$\begin{array}{c} L\left(X\right)=\sigma \left(B\left(PW{\mathrm{Conv}}_{2}\left(\delta \left(B\left({\mathrm{PWConv}}_{1}(X)\right)\right)\right)\right)\right) \end{array}$$
where PWConv, σ, δ, and *B* represent a 1×1 convolution, the sigmoid function, ReLU, and BN, respectively. The kernel sizes of $PWConv_{1}$ and $PWConv_{2}$ are (C/4)×C×1×1 and C×(C/4)×1×1, respectively. The attention weight map $L\left(X\right)\in R^{C^{\times }H^{\times }W}$ has the same shape as the feature map, so it can span channels spatially and highlight subtle details in an element-wise manner. Therefore, the bottom-up feature pyramid fusion module can dynamically perceive the subtle details of weak infrared targets.

Given *X* as the low-level feature and *Y* as the high-level feature, the cross-layer fusion feature $Z\in R^{C^{\times }H^{\times }W}$ can be obtained through the bottom-up feature pyramid fusion module. The calculation process can be expressed as(10)$$\begin{array}{c} Z=X+L_{X}⊗Y \end{array}$$
where ⊗ represents element multiplication.

### 3.5. SLS Loss Function

YOLOv7 uses the bounding box positioning loss function CIoU (Complete-IoU) [20]. CIoU’s design methodology integrates three geometric considerations: the overlapping area between the predicted box and the ground truth box, the center point distance, and the aspect ratio. Soft IoU [21] is also commonly used in semantic segmentation tasks. It is based on the traditional IoU indicator and also performs smoothing to make the prediction result smoother, which affects the calculation of the intersection-over-union ratio. However, for specific application scenarios where small targets usually only contain a few pixels, the evaluation indicators commonly used in these traditional target detection algorithms, such as Iou, SIou, DIou, and Soft IoU, lack sensitivity to the size and position of the target. Targets of different scales and positions may have the same IoU loss or Dice loss, which restricts the detection performance of small targets.

To resolve these limitations, we employ the scale and location sensitive (SLS) loss function, specifically designed to address the scale and positional insensitivity inherent in conventional loss formulations. It consists of scale-sensitive loss and position-sensitive loss and can be expressed as(11)$$\begin{array}{c} L_{SLS}=L_{S}+L_{L} \end{array}$$
where $L_{S}$ and $L_{L}$ represent scale-sensitive loss and position-sensitive loss; the scale-sensitive loss is implemented by providing weights through IoU loss. Let $A_{p}$ and $A_{gt}$ be the set of predicted and ground truth pixels of the target, respectively.(12)$$\begin{array}{c} L_{s}=1-\omega \frac{|A_{p}⋂A_{gt}|}{A_{p}⋃A_{gt}} \end{array}$$(13)$$\begin{array}{c} s.t.\omega =\frac{\min(\left|A_{p}\right|,\left|A_{gt}\right|)+Var(\left|A_{p}\right|,\left|A_{gt}\right|)}{\max(\left|A_{p}\right|,\left|A_{gt}\right|)+Var(\left|A_{p}\right|,\left|A_{gt}\right|)} \end{array}$$
where Var(·) is a function that obtains the variance of the provided scalar. It can be seen that the smaller ω is, the larger the gap between |Ap| and |Agt| is (assuming that the IoU between |Ap| and |Agt| is fixed), that is, the predicted scale and the true scale are very different. At this time, the detector should pay more attention to the target with the larger loss.

The position-sensitive loss is calculated based on the predicted center point and the ground truth center point. Given the sets of predicted pixels $A_{p}$ and ground truth pixels $A_{gt}$, the center points corresponding to $A_{p}$ and $A_{gt}$ are obtained by averaging the coordinates of all pixels, denoted as $c_{p}=\left(x_{p},y_{p}\right)$ and $c_{gt}=\left(x_{gt},y_{gt}\right)$, respectively. Then, the coordinates of these two center points are converted to polar coordinates. For example, for $c_{p}$, the corresponding distance $d_{p}$ and angle $\theta_{p}$ in polar coordinates are(14)$$\begin{array}{c} d_{p}=\sqrt{x_{p}^{2}+y_{p}^{2}} \end{array}$$(15)$$\begin{array}{c} \theta =\arctan(\frac{y_{p}}{x_{p}}) \end{array}$$(16)$$\begin{array}{c} L_{L}=(1-\frac{\min(d_{p},d_{gt})}{\max(d_{p},d_{gt})})+\frac{4}{\pi^{2}}{\left(\theta_{p}-\theta_{gt}\right)}^{2} \end{array}$$
where $d_{gt}$ and $\theta_{gt}$ are the distance and angle of $c_{gt}$ in the polar coordinate system, respectively.

## 4. Experiment

### 4.1. Experimental Data and Experimental Settings

Dataset. The datasets used for model evaluation are the publicly available infrared small target datasets, NUDT-SIRST and IRSTD-1k, for training and prediction. We set the training and testing ratio to 3 for the NUDT-SIRST dataset (i.e., 995 images for training and 332 images for testing, where total number of targets ($T_{all}$) is 451 and total number of target pixels ($P_{all}$) is 14,101). The image size of the NUDT-SIRST dataset is 256 × 256. For the IRSTD-1k dataset, the training and testing ratio is set to 4 (i.e., 800 images for training and 201 images for testing, where total number of targets ($T_{all}$) is 297 and total number of target pixels ($P_{all}$) is 14,534). Before training, all images are normalized. For the IRSTD-1k dataset, the image size is resized to 256 × 256 resolution before being input into the network.

Hyperparameter Configuration. In this paper, The U-net paradigm with ResNets [22] was chosen as our segmentation backbone, which generates pixel-level segmentation maps. The number of down-sampling layers i is set to 4 due to the fact that shallower U-Net variants are more effective in retaining detailed information in tasks that require high spatial accuracy [23,24,25]. The Adagrad [26] method is used to optimize the network. The network model was trained for a total of 399 epochs, with a batch size of 16 and a learning rate set to 0.05. The experiments in this paper were all conducted on the Ubuntu 18.04 operating system, equipped with an Intel Xeon E5-2678 CPU and an NVIDIA Geforce TITAN RTX 24 GB GPU. The computational environment utilized Python 3.8, and the deep learning models were developed using the PyTorch 5.1.2 framework. Moreover, the latencies of all models are tested on NVIDIA Orin Nano with TensorRT FP16 machines, which have a computational power of 40TOPS. The hyperparameter settings for the model-driven approach are shown in Table 1.

**Table 1 sensors-25-00814-t001:** Detailed hyper-parameter settings of model-driven methods for comparison.

Methods	Hyperparameter Settings
Top-hat [11]	Structure shape: square, local window size: 5 × 5
IPI [5]	Patch size: 50 × 50, stride: 10, $\lambda =L/\sqrt{\min(I,J,P)}$, L=3, threshold factor: $\varepsilon =10^{-7}$, k=10

### 4.2. Performance Evaluation

Object detection usually uses metrics such as IoU, precision, and recall. These pixel-level evaluation indicators focus more on target shape evaluation. However, since small infrared targets usually lack shape and texture, $P_{d}$ and $F_{a}$ must be combined to evaluate positioning performance. Therefore, we use the intersection over union (IoU) to evaluate the ability to describe the shape of a small target and the detection probability $P_{d}$ and false alarm rate $F_{a}$ to evaluate the ability to locate small targets.

IoU is a pixel-level evaluation that assesses the method’s ability to describe a profile. IoU is calculated by the ratio of the intersection and union areas between the prediction and the label and is defined as follows(17)$$\begin{array}{c} IoU=\frac{A_{inter}}{A_{union}} \end{array}$$
where $A_{inter}$ and $A_{union}$ represent the interaction area and the union area.

Detection probability ($P_{d}$) is an object-level evaluation metric. It measures the ratio of the correctly predicted target numbers $T_{correct}$ over all target numbers $T_{all}$. $P_{d}$ is defined as follows(18)$$\begin{array}{c} P_{d}=\frac{T_{correct}}{T_{all}} \end{array}$$

The false alarm rate ($F_{a}$) is another target-level evaluation metric. It is used to measure the ratio of falsely predicted pixels $P_{false}$ over all target pixels $P_{all}$. $F_{a}$ is defined as follows(19)$$\begin{array}{c} F_{a}=\frac{P_{false}}{P_{all}} \end{array}$$

### 4.3. Performance Comparison with Previous Methods

To further verify the effectiveness of our proposed IDNA-UNet on the infrared dim target detection dataset, we evaluate it with different types of methods from both quantitative and qualitative perspectives, including traditional methods and deep learning-based methods. Traditional methods are Top-hat and IPI; deep learning-based methods are MDvsFA, ALCNet [7], DNANet, and MSHNet. For a fair comparison, all evaluated deep learning-based models are retrained using the official code to converge on the NUDT-SIRST [27] and IRSTD-1k [28] datasets.

Table 2 lists the evaluation indicators of IoU (%), $P_{d}$ (%), and $F_{a}$ ($10^{-6}$) of the six different methods on the NUDT-SIRST and IRSTD-1k datasets. These results demonstrate that the IDNA-UNet model has an excellent effect on infrared dim small target detection. The IDNA-UNet target detection method proposed in this paper has a higher IoU and lower $F_{a}$ than the mainstream method. Compared with the better-performing DNANet, $F_{a}$ can be reduced by 11.11 × 106. Our method’s detection rate ($P_{d}$) has also achieved a higher accuracy. The ROC curve of the Figure 5 also highlights that our model quickly reaches the top-left corner and demonstrates competitive performance in terms of AUC.

Additionally, the table provides detailed information regarding the computational complexity and parameter counts for five state-of-the-art infrared small target detection networks, offering insights into their efficiency and resource requirements. From the perspective of lightweight design, MD&FA, ALCNet, and MSHNet exhibit lower computational complexity, while in terms of segmentation performance, DNANet, MSHNet, and the proposed IDNA-UNet demonstrate higher segmentation accuracy. Considering engineering applications, deployment tests on the NVIDIA Orin Nano embedded platform reveal that ALCNet performs best in real-time capability, whereas IDNA-UNet effectively balances segmentation accuracy and computational complexity, with its inference speed generally meeting real-time requirements.

Traditional algorithms such as Top-hat and IPI are easily disturbed by dense background noise, resulting in large false alarm areas, or even missed detection, with poor performance. In contrast, deep learning algorithms such as MD&FA have much better results, but they are also prone to identifying small false alarm areas at some local high points. They cannot complete high-quality multi-scale small target detection. IDNA-UNet can still accurately detect targets of different sizes in an environment with a low signal-to-noise ratio while effectively avoiding noise interference and reducing the false alarm rate.

We visualized the infrared weak small target results of each method, as shown in Figure 6. As can be seen from rows 1 and 2, when there are fewer targets or the background is relatively simple, all algorithms, including traditional target detection algorithms, have good performance. However, the Top-hat algorithm performs poorly in shape segmentation performance.

As can be seen from rows 3 and 4, when there is background interference of high brightness and suspected small targets around the target, the Top-hat and IPI algorithms cannot adapt to the needs of this scene. The probability of missed detection and false alarms is very high. Both MDvsFA and MSHNet have missed detections. The occurrence of false alarms shows that relying only on small-area target detection will increase the risk of false alarms, which highlights the effectiveness of our proposed method in integrating global and local features.

For row 5, which is more difficult to detect due to the complex background around it and more than one small target in the scene, the traditional detection algorithm and MD&FA have a large number of false alarms. In contrast, the MSHNet and DNANet algorithms have missed detections. IDNA-Net is suitable for complex scenarios and generates outputs with accurate shapes and low false positive rates. This further highlights that our method not only ensures the detection rate but also achieves a good balance between the detection rate and the false alarm rate.

### 4.4. Ablation Experiments

We apply ablation experiments to demonstrate the effectiveness of the proposed modules and techniques. Experiments without DNIM use a conventional residual network-based U-Net network. The experiment uses $P_{d}$, $F_{a}$, and IoU to evaluate the target localization ability and boundary prediction ability of our method.

In Table 3, compared with the conventional U-Net network, the IoU and Fa values of the DNIM backbone network are significantly reduced by 18.84% and 50.24%. Therefore, the feature extraction module has higher detection accuracy and a lower false alarm rate for small target detection.

The BFPFM module and SLS loss function have a certain improvement effect on the infrared target detection effect. If the BFPFM module is removed, the performance of IoU, Pd, and Fa values on the NUDT-SIRST dataset will decrease by 1.5%, 0.73%, and 10.94 ×$10^{-6}$. Therefore, a better feature fusion effect can be achieved through adaptive feature enhancement of the multi-scale head and BFPFM module.

The use of the SLS loss function has improved IoU, $P_{d}$, and $F_{a}$ values. In particular, the SLS loss function has the greatest impact on $F_{a}$. Suppose the SLS loss function is not used. In that case, Fa performance will drop by 13.33 ×$10^{-6}$, indicating that the loss function can guide the optimization of network parameters during training and improve the detection performance of small targets without affecting the network structure.

### 4.5. Comparison of SLS Loss Function Performance

We also compare the detection performance of mainstream target detection methods using SLS and Soft IoU loss functions. Table 4 shows the results of infrared dim target detection methods trained using different loss functions.

The detection performance and IoU indicators of different detectors have been improved through the SLS loss function, so the introduction of loss can effectively improve the small target detection performance of the IDNA-UNet model. In addition, through the ablation experiments of DNANet and MSHNet models, it can be seen that the improvement effects are different for different model architectures. In general, the SLS loss function has certain effectiveness and generalization in the field of small target detection.

## 5. Conclusions and Discussion

We propose a new method called IDNA-UNet for infrared weak and small target detection. We input the image into the feature extraction network, which is composed of densely nested interactive modules (DNIMs) based on U-Net. Dense connection paths are added to the U-Net structure to achieve level-by-level feature fusion, which is conducive to retaining the detailed information of the target in the feature extraction. The channel attention module SENet is introduced in the residual unit module of each DNIM to re-weight the feature map of each channel. In the feature fusion network, a bottom-up feature pyramid fusion module is designed to modulate multi-scale features, embed the small-scale information of low-level features into high-level semantic feature information, and further retain the small target features. From the detection results, this algorithm has a relatively superior performance. It can achieve an excellent balance between the detection rate and the false alarm rate while ensuring the detection rate, thereby completing the infrared small target detection task under different complex backgrounds. Although the algorithm in this paper has made some breakthroughs in infrared weak small target detection, it still faces challenges in practical applications, such as high computational burden and poor adaptability to small targets and dynamic backgrounds. These data-driven methods are all customized segmentation networks, which are essentially black-box models, rather than combining deep learning-based small target detection algorithms with traditional domain knowledge. Only a few studies have attempted such an approach [7,29,30]. In the future, further progress can be made by leveraging the efficient noise suppression and background modeling capabilities of traditional methods, along with the ability of deep learning to automatically extract useful features from complex data. A more proactive exploration of combining traditional image processing techniques with deep learning methods, in an almost “white-box” manner, can be undertaken for infrared small target detection tasks. This will help achieve more stable, accurate, and efficient detection when dealing with complex backgrounds, multi-target scenarios, and low signal-to-noise ratio challenges.

## Figures and Tables

**Figure 1 sensors-25-00814-f001:**
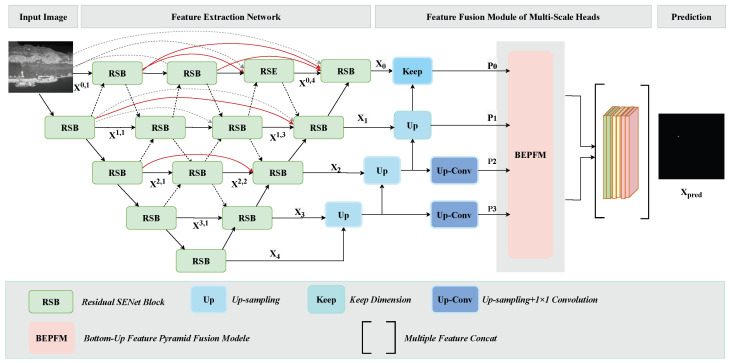
The architecture of IDNA-UNet consists of the feature extraction module and the feature fusion module.

**Figure 2 sensors-25-00814-f002:**
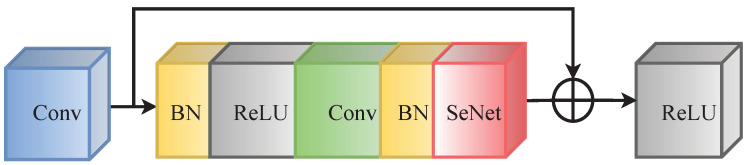
The network structure of convolutional neural network with residual connections.

**Figure 3 sensors-25-00814-f003:**
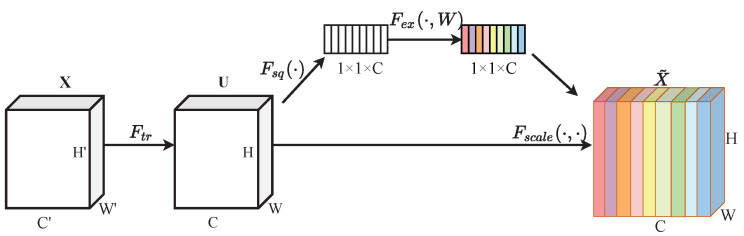
The structure of SENet, including squeeze and excitation.

**Figure 4 sensors-25-00814-f004:**
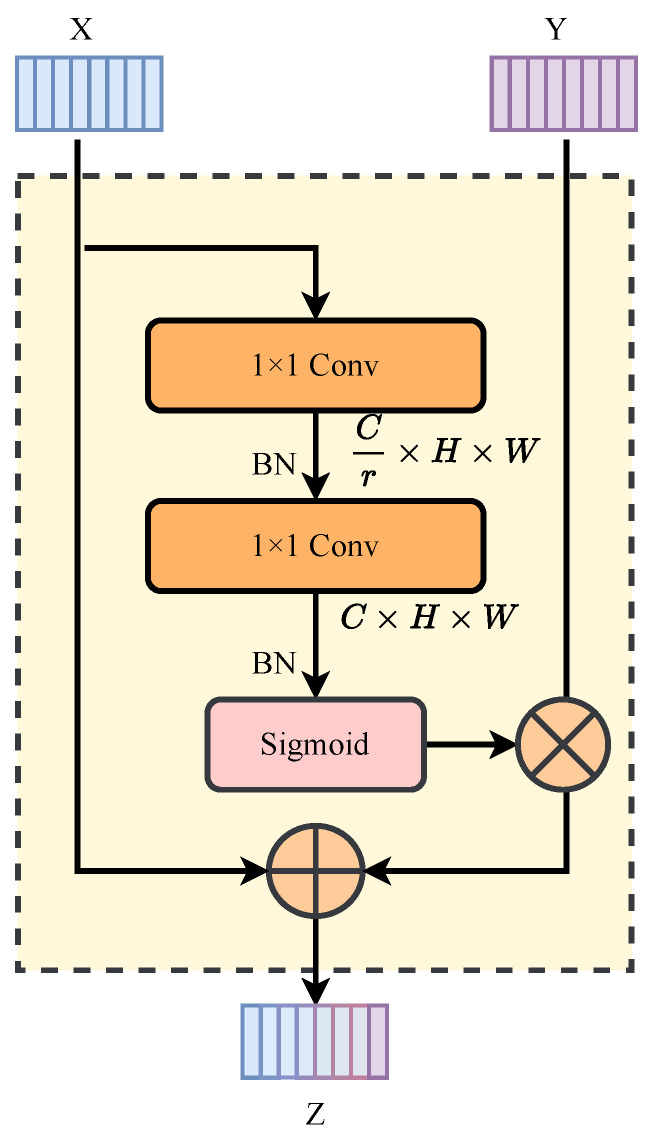
The structure of the bottom-up feature pyramid fusion module.

**Figure 5 sensors-25-00814-f005:**
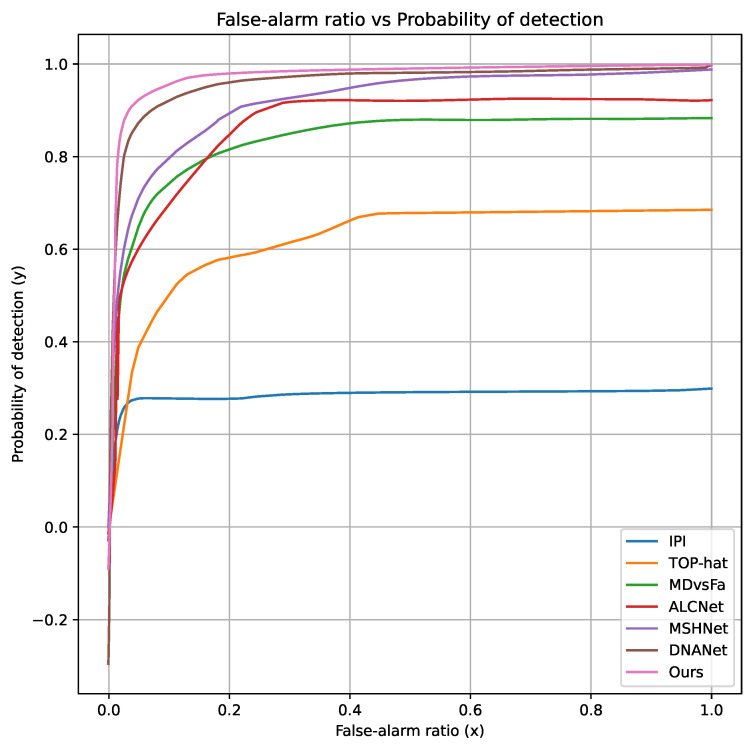
ROC curves of different methods on NUDT-SIRST.

**Figure 6 sensors-25-00814-f006:**
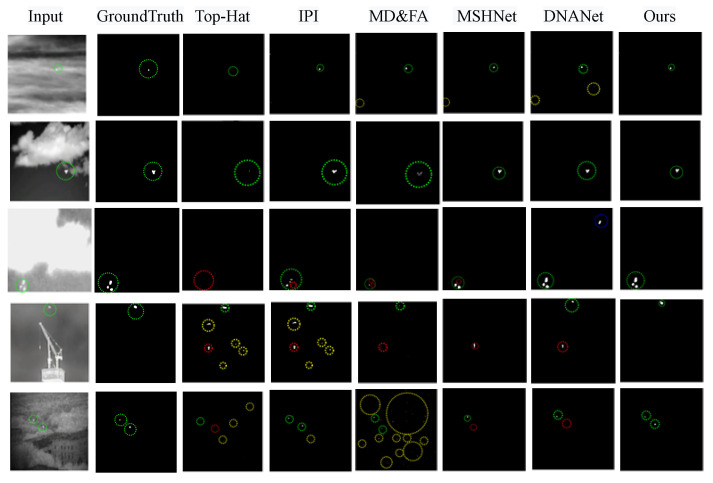
Visual comparison of experimental results of different methods. The targets to be detected are marked with green circles, and missed detections and false detections are marked with red and yellow dotted circles, respectively.

**Table 2 sensors-25-00814-t002:** Experimental results on the NUDT-SIRST and IRSTD-1k dataset.

Model	NUDT-SIRST			ISTD-1k			#Params (M)	Latency (ms)
	IoU↑	*Pd*↑	*Fa*↓	IoU↑	*Pd*↑	*Fa*↓		
Top-hat	22.34	70.54	95.37	12.56	68.94	188.78	-	-
IPI	18.67	67.87	45.78	26.01	70.39	36.69	-	-
MD&FA	52.84	82.34	50.55	52.88	86.9	31.32	3.13	9.66
ALCNet	78.45	96.34	35.99	48.02	93.3	29.87	1.5	6.63
DNANet	82.17	98.93	23.65	63.01	97.7	9.717	4.69	24.05
MSHNet	81.07	98.25	25.81	64.51	97.8	12.37	4.05	9.06
Ours	82.34	98.75	12.54	66.15	97.8	7.1	4.68	17.68

**Table 3 sensors-25-00814-t003:** Ablation of the DNIM, BFPFM, and $L_{SLS}$ in IDNA-UNet.

DNIM	BFPFM	$L_{\mathrm{SLS}}$	IoU↑	$P_{d}↑$	$F_{a}↓$
×	✓	✓	63.50	97.70	62.78
✓	×	✓	80.84	98.02	23.48
✓	✓	×	82.27	98.01	25.87
✓	✓	✓	82.34	98.75	12.54

**Table 4 sensors-25-00814-t004:** Performance comparison of common object detection methods (IDNA-UNet, DNANet, MSHNet) using LSLS and Soft IoU loss functions.

Loss Function	IDNA-UNet			DNANet			MSHNet		
	IoU↑	$P_{d}↑$	$F_{a}↓$	IoU↑	$P_{d}↑$	$F_{a}↓$	IoU↑	$P_{d}↑$	$F_{a}↓$
$L_{iou}$	82.27	98.01	25.87	82.17	96.93	23.65	81.03	97.80	24.66
$L_{SLS}$	82.34	98.75	12.54	80.75	97.31	41.13	81.07	98.25	25.81

## Data Availability

The data used to support the results of this study are included in the article.

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
