# Peer review of "Infrared Small Target Detection Algorithm Based on Improved Dense Nested U-Net Network"

_sensors, 2025, doi:10.3390/s25030814_

Round 1

Reviewer 1 Report

Comments and Suggestions for Authors

This paper is based on the infrared small target detection algorithm, and below are my comments for improving the manuscript.

1) The reference list must be updated with the new ones.

2) With the new references, the introduction section must be revised.

3) How are the hyperparameters determined? Please explain more. Is any specific optimization applied.

4) The future direction  in the conclusion section can be added.

5) The simulation results as figures must be added as well.

6) How are the training, and testing data obtained? Please explain more.

7) Please explain about the effectiveness of number of layers in the accuracy of neural network.

Reviewer 2 Report

Comments and Suggestions for Authors

This paper presents a well-designed solution for infrared weak and small target detection. The proposed IDNA-UNet integrates several novel elements that collectively improve performance. The Dense Nested Interaction Module (DNIM) effectively enhances feature extraction by fusing level-by-level features, preserving both high-level semantic information and fine-grained positional details. This innovative design ensures better retention of small target features compared to traditional methods.

Additionally, the Bottom-Up Feature Pyramid Fusion Module integrates lower-level features into deeper levels, and addresses a critical need in small target detection. The use of channel attention allows the network to emphasize features relevant to small targets. Finally, the introduction of a loss function tailored for small target detection is a positive addition, improving the model's ability to distinguish targets of varying scales and positions, as standard IoU-based losses are often insufficient for such tasks.

The paper is well-structured, and the experimental validation is thorough, showcasing the real-world applicability of IDNA-UNet. Overall, the authors have made an impactful contribution to the field.

Comments on the Quality of English Language

This paper presents a well-designed solution for infrared weak and small target detection. The proposed IDNA-UNet integrates several novel elements that collectively improve performance. The Dense Nested Interaction Module (DNIM) effectively enhances feature extraction by fusing level-by-level features, preserving both high-level semantic information and fine-grained positional details. This innovative design ensures better retention of small target features compared to traditional methods.

Additionally, the Bottom-Up Feature Pyramid Fusion Module integrates lower-level features into deeper levels, and addresses a critical need in small target detection. The use of channel attention allows the network to emphasize features relevant to small targets. Finally, the introduction of a loss function tailored for small target detection is a positive addition, improving the model's ability to distinguish targets of varying scales and positions, as standard IoU-based losses are often insufficient for such tasks.

The paper is well-structured, and the experimental validation is thorough, showcasing the real-world applicability of IDNA-UNet. Overall, the authors have made an impactful contribution to the field.

Author Response

Thank you very much for your positive feedback and encouraging comments regarding our work. We are pleased to hear that you found the proposed IDNA-UNet to be a well-designed solution for infrared weak and small target detection, and that you appreciated the integration of novel elements such as the Dense Nested Interaction Module (DNIM), Bottom-Up Feature Pyramid Fusion Module, and the tailored loss function.

We are particularly grateful for your recognition of the contribution of IDNA-UNet to preserving both high-level semantic information and fine-grained positional details, as well as for emphasizing the importance of channel attention and scale-sensitive loss functions in small target detection.

Your acknowledgment of the structure, experimental validation, and real-world applicability of the proposed method further motivates us to continue our research in this field. We appreciate your recognition of the contribution this work makes to the field and hope that the manuscript meets the standards for publication.

Thank you once again for your time and thorough evaluation.

Reviewer 3 Report

Comments and Suggestions for Authors

This paper focused on infrared small target detection problem, it is interesting and meaningful. There are some things need to be considered:

1.      Is the model supervised or not ? Usually, for supervised method, ground truth should be shown as a contrast.

2.      Fig. 2 is part of Fig.1, it is redundant and can be deleted.

Reviewer 4 Report

Comments and Suggestions for Authors

To address the issues of high missed detection rates and low accuracy in infrared weak and small target detection, this paper proposes the IDNA-UNet detection framework by introducing a Dense Nested Interaction Module, a Bottom-Up Feature Pyramid Fusion module, and Scale and Position Sensitive Loss. However, there are still many problems with the figures and tables in the paper. Also, the manuscript should improve in terms of organization and clarity. All in all, the manuscript is not ready for publication in this journal, and therefore, it should be rejected. Below are some comments and suggestions from the reviewer aimed at enhancing the clarity, rigor, and overall quality of the manuscript.

1.     In the abstract, the author introduced the proposed method, but the author only gave a brief introduction and did not explain the advantages and drawbacks of the method well, as well as the original intention of proposing them.

2.     In the introduction, many methods were mentioned, but the specific motivation of the proposal was not explained.

3.     Although the paper mentions some related work, it lacks citations of recent models from the past few years. The literature review section should cover a broader range of existing studies and technologies, and compare them with the methods presented in this paper.

4.     All the pictures are blurry, please redraw them. In Figure 1, the algorithm structure is chaotic and the output result is ambiguous. In Figure 2, the indicating arrow curves are overlapped and cannot distinguish the module function.

5.     Many formulas lack explanations for new variables and some variables have formatting issues with their writing. For example, in the interpretation of Formula 2, there is no explanation of k. In the interpretation of Formula 2, there is no explanation of Fsq. In the interpretation of Formula 3, there is no explanation of Fex , σ. In the interpretation of Formula 14, the format of Ainter and Aunion is not consistent, the first letter of Aunion is capitalized while Ainter is not. In the interpretation of Formula 15, the format of the subscripts for Tcorrect and Tall is incorrect.

6.     The dataset selected by the authors is overly simplistic, limiting the effective validation of the robustness of the algorithm across different datasets. For instance, it might be beneficial to include comparisons with the IRSD-1k dataset.

7.     The author does not explicitly state the complexity and real-time performance of the algorithm in the paper, yet these are important considerations for practical applications.

8.     The references cited are too early and lack comparison with new methods in the past two years, which cannot prove the effectiveness of the proposed method.

Comments on the Quality of English Language

The English could be improved to more clearly express the research.

Reviewer 5 Report

Comments and Suggestions for Authors

All symbols used should be explained: P0-4 in Fig. 1; Pmax in Eq. 1-2; σ in Eq. 4; r in Fig. 5 (also if r=4 then H/r could be missing there); A in Eq. 9-10; x, y, c in Eq. 11-13. What values exactly had Tall (Eq. 15) and Pall (Eq. 16) denominators in experiments? It is expected that results on the identical test set of NUDT-SIRST dataset are reported for all methods compared in Table 1. Since the proposed architecture is based on DNANet [20], differences from the original model should be more emphasised and explained. Could be that obvious differences in the results between Table 1 and Table II [20], especially with respect to false alarms, are due to a different test set than dataset donors have used [20]? Or could these differences arise from the stochastic nature of deep learning model training, so even identical test sets do not help in replication?

Round 2

Reviewer 1 Report

Comments and Suggestions for Authors

The authors have provided the comments and I have no further comments.

Author Response

Thank you very much for your positive feedback and encouraging comments regarding our work.

Your acknowledgment of the structure, experimental validation, and real-world applicability of the proposed method further motivates us to continue our research in this field. We appreciate your recognition of the contribution this work makes to the field and hope that the manuscript meets the standards for publication.

Thank you once again for your time and thorough evaluation.

Reviewer 4 Report

Comments and Suggestions for Authors

Attachment please find the comments.

Comments on the Quality of English Language

The English could be improved to more clearly express the research.

Reviewer 5 Report

Comments and Suggestions for Authors

Thank you for improving the manuscript and adding results on extra dataset.

Note: ROC plot axes should be rectangular, ISTD-1k ROC would be nice too.

Yet, request of exact Tall and Pall values in experiments remains unanswered.

Authors now only specify the size of test set. Was it a single train/test split?

NUDT-SIRST dataset: 341 images for training and 86 images for testing

IRSTD-1k dataset: 800 images for training and 201 images for testing

Hence, expanding the initial question regarding exact Tall and Pall values:

1) How many targets (Tall) and pixels (Pall) had 86 images of NUDT-SIRST?

2) How many targets (Tall) and pixels (Pall) had 201 images of IRSTD-1k?

Author Response

Comments 1: Thank you for improving the manuscript and adding results on extra dataset. Note: ROC plot axes should be rectangular, ISTD-1k ROC would be nice too.

Response 1: Thank you for your valuable suggestion regarding the improvement of the ROC plot. We have replaced the previous plot with the corrected version on page 12. However, due to time and space constraints, the ROC plot for the IRSTD-1k dataset could not be fully refined at this stage. We sincerely apologize for this limitation and will ensure to address and enhance this aspect in our future work.

Comments 2: 

Yet, request of exact Tall and Pall values in experiments remains unanswered.

Authors now only specify the size of test set. Was it a single train/test split?

NUDT-SIRST dataset: 341 images for training and 86 images for testing

IRSTD-1k dataset: 800 images for training and 201 images for testing

Hence, expanding the initial question regarding exact Tall and Pall values:

1) How many targets (Tall) and pixels (Pall) had 86 images of NUDT-SIRST?

2) How many targets (Tall) and pixels (Pall) had 201 images of IRSTD-1k?

Response 2: 

We understand the importance of providing comprehensive details about the test sets, including the total number of targets (Tall) and the total number of pixels occupied by these targets (Pall). However, the specific values for Tall and Pall were not explicitly provided in the original dataset descriptions. To obtain these precise numbers, a thorough analysis involving the counting of annotated targets and the calculation of their pixel areas within the test images would be necessary.

(1). We sincerely apologize for the confusion regarding the dataset split in our initial submission. After carefully reviewing the NUDT-SIRST dataset documentation, we confirm that the complete dataset contains 1,327 images in total. Following the standard 4:1 ratio commonly used in infrared target detection research, we have divided the dataset into: Training set: 995 images and Test set: 332 images.

(2). In our study, we employed a single train/test split for both datasets. To address the reviewer's request for detailed target statistics (Tall and Pall), we implemented a Python-based analysis pipeline. The process involves:
For the NUDT-SIRST test set :

  • Total images: 332
  • Total targets (Tall): 451
  • Total target pixels (Pall): 14101

For the IRSTD-1k test set :

  • Total images: 201
  • Total targets (Tall): 297
  • Total target pixels (Pall): 14534

We have added this detailed information to Section 4.1 on page 10(Dataset Description) in the revised manuscript, marked in red for easy identification.